# DFCA: Decentralized Federated Clustering Algorithm

## Abstract

Clustered Federated Learning has emerged as an effective approach for handling heterogeneous data across clients by partitioning them into clusters with similar or identical data distributions. However, most existing methods, including the Iterative Federated Clustering Algorithm (IFCA), rely on a central server to coordinate model updates, which creates a bottleneck and a single point of failure, limiting their applicability in more realistic decentralized learning settings. In this work, we introduce DFCA, a fully decentralized clustered FL algorithm that enables clients to collaboratively train cluster-specific models without central coordination. DFCA uses a sequential running average to aggregate models from neighbors as updates arrive, providing a communication-efficient alternative to batch aggregation while maintaining clustering performance. Our experiments on various datasets demonstrate that DFCA outperforms other decentralized algorithms and performs comparably to centralized IFCA, even under sparse connectivity, highlighting its robustness and practicality for dynamic real-world decentralized networks.[1]

## 1 Introduction

Federated Learning (FL) has emerged as a new paradigm that allows for clients to train Machine Learning (ML) models collaboratively without the need to share their raw data. By enabling collaborative training across multiple devices, FL has gained significant attention in research and industry, especially since distributed computing with different devices has become a crucial component of modern technology. The most known FL implementation strategy, FedAvg (McMahan et al., 2017), and most other known FL algorithms assume a setting with a central instance that aggregates the local updates of all clients to form a global model, which is then broadcast back to the network. While effective, this orchestration with a central server introduces several limitations, including a single point of failure, communication delays, and bottlenecks that are often connected to more challenging learning settings with Internet of Things (IoT) devices and mobile phones (Lalitha et al., 2018).

To address the limitations of centralized Federated Learning (CFL), recent research has explored decentralized Federated Learning (DFL), where clients communicate with each other without the need for a central instance (Lalitha et al., 2018). Decentralized strategies often utilize peer-to-peer (P2P) (Lalitha et al., 2019) or gossip-based (Hu et al., 2019; Hegedűs et al., 2019) exchange methods to achieve convergence through direct communication between clients. DFL approaches remove the single point of failure, often reduce communication cost and delays, and improve the overall robustness (Yuan et al., 2024).

Concurrently, clustered FL has appeared as a proposed solution to data heterogeneity across clients, another major issue in ML and FL. In most real-world scenarios, the data is not independently identically distributed (non-IID) over all clients, making global aggregation less efficient and suboptimal. Clustered FL methods attempt to cluster clients into groups with similar data distributions, allowing clusters to capture local patterns and characteristics during training (Sattler et al., 2019). The most popular among clustered FL techniques is the Iterative Federated Clustering Algorithm (IFCA) (Ghosh et al., 2021), which is a centralized, training loss-based clustering method, where clients clusters are evaluated locally after each global training round. As most other clustered FL

---

[1]The code is attached to the submission and will be made publicly available upon acceptance.

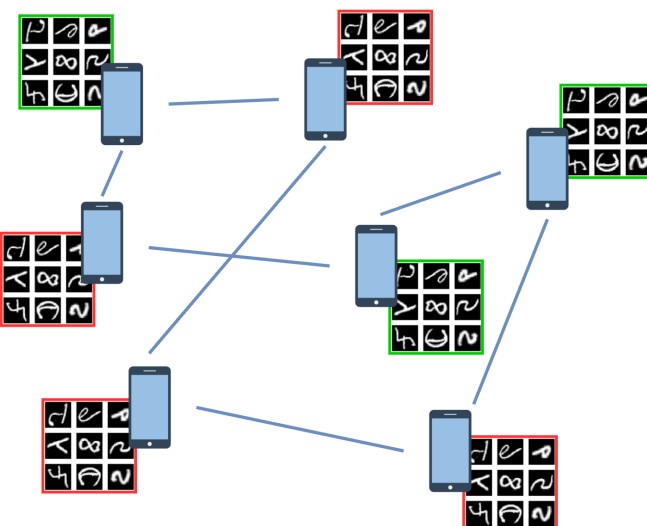

Figure 1: Illustration of the DFCA problem for Rotated EMNIST with two different data distributions

techniques that have been developed over the recent years also presume central instance coordination, they are not optimized for decentralized learning settings. In this paper, we propose in this paper the Decentralized Federated Clustering Algorithm (DFCA) to address this issue.

**Our contributions:**

1. We formulate DFCA, a fully decentralized federated clustering algorithm inspired by IFCA and designed to operate effectively in low-connectivity networks with heterogeneous client data distributions.

2. We incorporate a sequential running-average parameter exchange strategy that preserves clustering performance while enabling communication-efficient updates across the network.

3. Through extensive experiments on various datasets, we demonstrate that DFCA matches the accuracy of the centralized IFCA baseline and outperforms decentralized alternatives. Furthermore, sequential aggregation achieves performance comparable to synchronous batch aggregation, highlighting its practicality for real-world decentralized settings.

After looking at the problem formulation in Section 2, we proceed to introduce our method in Section 3, analyze its convergence in Section 4, and show our simulation results in Section 5. We will conclude this paper's findings in Section 7, after briefly introducing related work in Section 6.

## 2 PRELIMINARIES

Let $M$ be a set of $N$ clients that are connected to each other in a graph. We represent the graph by $N$ sets $\mathcal{N}_i \subset M$, which contain the neighboring clients for each client $i$ (i.e., neighborhood sets). The clients are partitioned into $k$ disjoint clusters $\mathcal{S}_1, ..., \mathcal{S}_k \subset M$. Each cluster is associated with a distinct data distribution $\mathcal{D}_1, ..., \mathcal{D}_k$. Our problem setup is illustrated in Figure 1, with the different data distributions being simulated by handwritten character digits (EMNIST) rotated by 0, 90, 180, 270 degrees.

For each client $i$, we sample a data set $D_i$ distributed according to $\mathcal{D}_j$ of the associated cluster $j$ meaning that each clients has data from one of $k$ data distribution. Additionally, at each client $i$ we store all $k$ machine learning models (ML-models), which are parameterized by $\theta_{i,j}$ where $j \in [k]$ and $k$ is the number of clusters. Client $i$ will update the parameters $\theta_{i,j}$ of the model, which is associated with its corresponding cluster $j$, by gradient descent using $D_i$. During aggregation (com-

munication phase) the local models for all clusters are updated using the models of the neighboring clients. Note that the corresponding (assigned) cluster of a client might change after an iteration.

For client-local learning, we consider a loss function $\mathcal{L}(\theta_{i,j}, d)$ that calculates the loss for a single data point $d \in D_i$. These losses can be combined on the client-level and also on the cluster-level:

Let's first consider the loss for an individual client. We assume that client $i$ is assigned to cluster $j$. Then we write the client-specific objective as

$$F_{\text{client}}(\theta_{i,j}, D_i) = \frac{1}{|D_i|} \sum_{d \in D_i} \mathcal{L}(\theta_{i,j}, d). \tag{1}$$

Second, we define the loss for each cluster $j$ as the sum of the losses of the associated clients,

$$F_{\text{cluster}}(j) = \sum_{i \in \mathcal{S}_j} F_{\text{client}}(\theta_{i,j}, D_i). \tag{2}$$

Finally, we define the global loss, which combines all data points across all clients into a single number:

$$F_{\text{global}} = \sum_{j=1}^{k} F_{\text{cluster}}(j) \tag{3}$$

Having formulated the loss functions on client- and cluster-level, we next introduce the decentralized learning algorithm DFCA, which allows clients to collaboratively minimize their respective cluster-specific losses while communicating with their neighbors in the graph to exchange results.

## 3 DECENTRALIZED FEDERATED CLUSTERING ALGORITHM

Decentralized Federated Clustering Algorithm (DFCA) starts with initialization of the model parameters and then iterates three steps: (1) Cluster Assignment, (2) Local Updates, and (3) Decentralized Aggregation. Steps (1) and (2) are similar to existing clustered FL algorithms (Ghosh et al. (2021); Lin et al. (2025); El-Rifai et al. (2025)). Step (3) enables decentralized learning.

**Initialization.** Before detailing the three iterative steps, we explain how the model parameters $\theta_{i,j}$ are initialized. We consider two variants: (i) with the *global initialization* method (DFCA-GI), all $k$ models are centrally generated and published via broadcast (or initialized locally using the same seed) before the first iteration, so that every client holds the same model parameters at the beginning. (ii) For the *local initialization* method (DFCA-LI), all clients start on different parameters, i.e., each client can initialize the models locally.

### 3.1 CLUSTER ASSIGNMENT

Every client $i$ is assigned to cluster $c(i) \in [k]$ through inference on the current parameters $\theta_{i,j}$. More formally, we update $c(i)$ to be the argmin of the local client loss,

$$c(i) \leftarrow \arg\min_{j} F_{\text{client}}(\theta_{i,j}, D_i) \tag{4}$$

Hereby, the overall loss $F_{\text{global}}$ is non-increasing. These cluster assignments are repeated at the start of each training loop.

### 3.2 LOCAL UPDATE

The local update at client $i$ runs several epochs at the client-level using (stochastic) gradient descent on the local data $D_i$ with respect to $\theta_{i,c(i)}$:

$$\theta_{i,c(i)} \leftarrow \theta_{i,c(i)} - \gamma \nabla F_{\text{client}}(\theta_{i,c(i)}, D_i) \tag{5}$$

(with learning rate $\gamma$), i.e., we only modify the parameters of the assigned cluster $c(i)$. Again, the gradient descent ensures that the global loss $F_{\text{global}}$ is decreasing (at least in expectation).

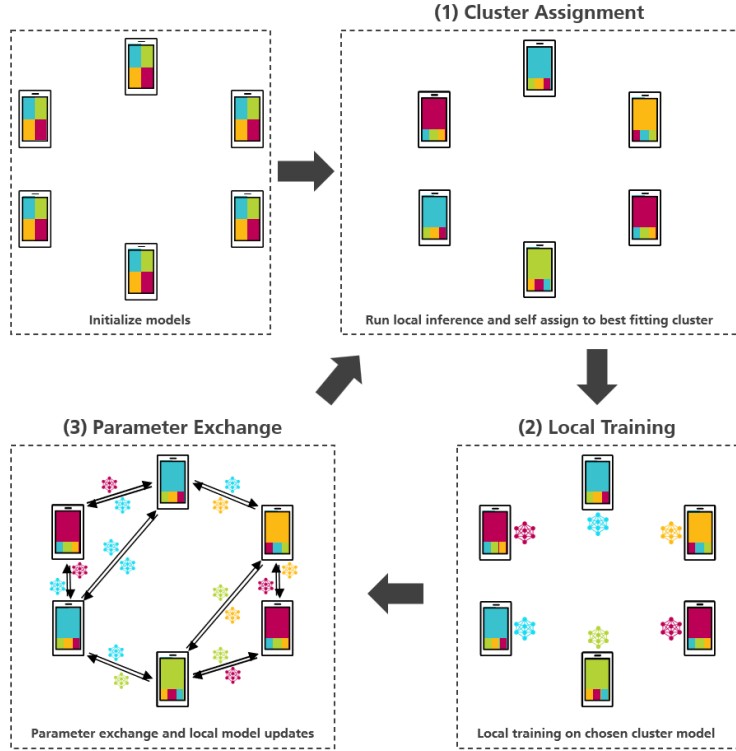

Figure 2: After initialization, DFCA iterates three steps: (1) cluster assignment, (2) local training, and (3) parameter exchange

### 3.3 DECENTRALIZED AGGREGATION (AKA COMMUNICATION STEP)

The goal of our algorithm is that at the end, all clients hold *all* $k$ trained models. Thus, limiting the communication to neighbors within the *same* cluster would be suboptimal. Instead, all clients exchange their parameters with *all* of their neighbors according to the graph and locally average the models. More formally, client $i \in M$ receives parameters from all neighbors in its neighborhood $\mathcal{N}_i$. To maintain cluster-specific updates in a sparse decentralized network, clients receive models from their neighbors but only send out the model parameters $\theta_{i,c(i)}$ that they trained themselves in the previous step.

To specify the aggregation equations, we split the neighbors of client $i$ according to their cluster assignments:

$$\mathcal{N}_{i,j} := \{m \mid m \in \mathcal{N}_i \text{ and } c(m) = j\} \subset \mathcal{N}_i \tag{6}$$

(for $i \in [N]$ and $j \in [k]$). In this phase, the clients update the parameter sets for all clusters, not only the one of their assigned cluster $c(i)$.

**Batch aggregation.** Next, we define the batch update (synchronous), which assumes that all neighbors $m$ have reported their current models $\theta_{m,j}$:

$$\theta_{i,j} \leftarrow \frac{1}{|\mathcal{N}_{i,j}| + 1} \left( \theta_{i,j} + \sum_{m \in \mathcal{N}_{i,j}} \theta_{m,j} \right) \tag{7}$$

(for $i \in [N]$ and $j \in [k]$).

**Sequential aggregation.** While batch aggregation is the perfect scenario, in practice, neighbors might report their updates asynchronously, and we can never be sure whether a client has disconnected or not. Thus we need sequential averaging that is robust against failing clients and random

arrival times. The basic idea is to replace the averaging in Eq. 7 with an online version (a.k.a. running average): we start with the local parameter value $\theta_{i,j}$ and update it as the messages from the other clients come in. Assuming $r$ neighbors have already reported their updates for cluster $j$, we update $\theta_{i,j}$ with:

$$\theta_{i,j} \leftarrow \frac{r}{r+1}\theta_{i,j} + \frac{1}{r+1}\theta_{m,j} \qquad \text{for } r \in [|\mathcal{N}_{i,j}|] \tag{8}$$

(for $i \in [N]$ and $j \in [k]$).

Our sequential aggregation naturally supports asynchronous updates, allowing each client to integrate neighbor models immediately as they arrive, which can improve efficiency and reduce idle time in fully distributed deployments. This approach is also memory efficient, as it only requires storing the current estimate per model rather than all neighbor updates. Moreover, using a running average ensures that each incoming model contributes proportionally to the aggregated model, providing a stable and principled approximation of the full batch aggregation even in dynamic and sparse networks.

---

**Algorithm 1** Decentralized Federated Clustering Algorithm (DFCA)

---

1: **Input:** number of clusters $k$, number of iterations $T$
2: **Local:** step size $\gamma$, number of local epochs $\tau$
3:
4: DFCA-GI: initialize $\theta_{i,j}$ per cluster and publish models to all clients
5: DFCA-LI: initialize $\theta_{i,j}$ for all clusters per client (personalized models)
6:
7: **for** $t = 0, 1, ..., T-1$ **do**
8:      $M_t \leftarrow$ subset of worker machines (participating devices)
9:      **for** worker machine $i \in M_t$ **do**
10:
11:          **Step 1:** AssignCluster
12:          $c(i) \leftarrow \arg\min_j F_{\text{client}}(\theta_{i,j}, D_i)$             ▷ run local inference on all models
13:
14:          **Step 2:** LocalUpdate
15:          **for** $q = 0, ..., T-1$ **do**
16:              $\theta_{i,c(i)} \leftarrow \theta_{i,c(i)} - \gamma\nabla F_{\text{client}}(\theta_{i,c(i)}, D_i)$      ▷ stochastic gradient descent
17:          **end for**
18:
19:          **Step 3:** Aggregation
20:          **for** each cluster $j = 1, ..., k$ **do**
21:              $r \leftarrow 0$
22:              **for** each neighbor $m \in \mathcal{N}_{i,j}$ **do**
23:                  $r \leftarrow r + 1$
24:                  $\theta_{i,j} \leftarrow \frac{r}{r+1}\theta_{i,j} + \frac{1}{r+1}\theta_{m,j}$      ▷ running average for each cluster
25:              **end for**
26:          **end for**
27:      **end for**
28: **end for**

---

## 4    CONVERGENCE SUMMARY

We briefly summarize the convergence properties of DFCA. Full proofs are deferred to Appendix B.

**Setup.** Each client stores all $k$ models $\{\theta_{i,j}^t\}$ here with index $t$ for the round. Each round executes three steps: (i) cluster assignment by local inference, (ii) local stochastic gradient descent on the assigned model, and (iii) decentralized aggregation with neighbors. Aggregation is carried out via gossip (either synchronous averaging or sequential running averages), which preserves the network-wide average and contracts disagreement among clients. For cluster $j$, define the stacked vector $\Theta_j^t = (\theta_{1,j}^t, \ldots, \theta_{N,j}^t)$ and the network average $\bar{\theta}_j^t = \frac{1}{N}\sum_{i=1}^N \theta_{i,j}^t$. We measure per-cluster

*disagreement* by $\mathrm{Disp}_j^t = \frac{1}{N} \sum_{i=1}^N \|\theta_{i,j}^t - \bar{\theta}_j^t\|^2$. The three steps of one DFCA update round can be written as:

1. **Assignment:** $c_t(i) = \arg\min_{j\in[k]} F_{\mathrm{client}}(\theta_{i,j}^t, D_i)$.

2. **Local descent (assigned index only):**

$$\theta_{i,c_t(i)}^{t+\frac{1}{2}} = \theta_{i,c_t(i)}^t - \gamma\, g_{i,c_t(i)}(\theta_{i,c_t(i)}^t), \qquad \theta_{i,j}^{t+\frac{1}{2}} = \theta_{i,j}^t \ \ (j \neq c_t(i)), \tag{9}$$

   with stochastic gradient $g_{i,j}$.

3. **Decentralized aggregation (all $j$):**

$$\theta_{i,j}^{t+1} = \sum_{m\in\{i\}\cup\mathcal{N}_i} w_{im,t}^{(j)}\, \theta_{m,j}^{t+\frac{1}{2}}, \tag{10}$$

   where $W_t^{(j)} = (w_{im,t}^{(j)})$ respects $G$, is row-stochastic, and is doubly-stochastic in the synchronous (batch) case. In the sequential/async case, $W_t^{(j)}$ is time-varying with standard joint-connectivity.

We adopt the following standard assumptions.

(A1) **Smoothness.** For all $i, j$, $F_{\mathrm{client}}(\cdot, D_i)$ is $L$-smooth.

(A2) **Noise.** Unbiased stochastic gradients with bounded variance: $\mathbb{E}[g_{i,j}(\theta) \mid \theta] = \nabla F_{\mathrm{client}}(\theta, D_i)$, $\mathbb{E}\|g_{i,j}(\theta) - \nabla F_{\mathrm{client}}(\theta, D_i)\|^2 \leq \sigma^2$.

(A3) **Graph mixing.** In the synchronous case there exists a symmetric, doubly-stochastic $W$ respecting $G$ with spectral gap $1 - \lambda > 0$ such that $\|XW - \mathbf{1}\bar{x}^\top\| \leq \lambda \|X - \mathbf{1}\bar{x}^\top\|$ for any row-stacked $X$. In the async case, $\{W_t^{(j)}\}$ are row-stochastic, edges are repeatedly activated with bounded delays, and there exists a window $B$ and $\tilde{\lambda} \in (0,1)$ such that over any $B$ consecutive rounds disagreement contracts by $\tilde{\lambda}$.

(A4) **Objective curvature.** Either (PL) each $F_{\mathrm{cluster}}(j; \cdot)$ satisfies the $\mu$-Polyak–Łojasiewicz (PL) inequality, or (Cvx) each is convex.

(A5) **Separability (IFCA-style).** There exists $\delta > 0$ such that, in a neighborhood of the cluster minimizers $\{\theta_j^\star\}_{j=1}^k$, the argmin-of-loss assignment selects the true cluster:

$$\mathbb{E}_{d\sim\mathcal{D}_{c(i)}}\big[\mathcal{L}(\theta_{i,c(i)}, d)\big] \leq \min_{j\neq c(i)} \mathbb{E}_{d\sim\mathcal{D}_{c(i)}}\big[\mathcal{L}(\theta_{i,j}, d)\big] - \delta. \tag{11}$$

On a high-level, the analysis combines two ingredients:

1. *Cluster assignment:* Choosing the best-fitting model index per client never increases the global loss, and after sufficient descent the assignments stabilize to the ground-truth clusters.

2. *Local descent + gossip:* Gradient descent decreases the cluster objectives, up to stochastic noise and a *disagreement penalty*. Gossip averaging preserves the average model and contracts disagreement at a rate governed by the graph spectral gap.

Together, these steps imply that DFCA behaves like $k$ independent instances of decentralized SGD, one per cluster, after a finite burn-in.

**Theorem 1** (Convergence of DFCA). *Assume (A1)–(A5), choose $\gamma \leq c/L$ for a small numerical constant $c$, and let $\lambda$ (resp. $\tilde{\lambda}$) be the consensus factor in the synchronous (resp. async) case. Then:*

(i) *(Pre-stabilization) $F_{global}^t$ is non-increasing in expectation across assignment and local steps. The disagreements $\{\mathrm{Disp}_j^t\}$ remain bounded and contract at rate $\lambda$ (or $\tilde{\lambda}$ over windows).*

(ii) *(Stabilization) There exists $\tau < \infty$ such that $c_t(i) = c_\star(i)$ for all $t \geq \tau$.*

(iii) *(Post-stabilization) For $t \geq \tau$, DFCA is $k$ independent copies of decentralized SGD on $F_{cluster}(j)$.*

- *Under (PL),*

$$\mathbb{E}\left[F_{global}^{\tau+T} - F_{global}^{\star}\right] \leq (1 - \mu\gamma/2)^T C_0 + O\left(\frac{\gamma\sigma^2}{\mu}\right) + O\left(\frac{\gamma L}{1-\lambda}\sigma^2\right), \quad (12)$$

  *with $C_0$ depending on the gap at $t = \tau$; in async, replace $(1 - \lambda)$ by the windowed $(1 - \tilde{\lambda})$.*

- *Under (Cvx),*

$$\frac{1}{T}\sum_{t=\tau}^{\tau+T-1}\sum_{j=1}^{k}\mathbb{E}\|\nabla F_{cluster}(j; \bar{\theta}_j^t)\|^2 \leq O\left(\frac{F_{global}^{\tau} - F_{global}^{\star}}{\gamma T}\right) + O(\gamma L\sigma^2) + O\left(\frac{\gamma L}{1-\lambda}\sigma^2\right),$$

$$(13)$$

  *and choosing $\gamma = \Theta(1/\sqrt{T})$ yields the usual $O(1/\sqrt{T})$ rates (with the consensus penalty).*

**Takeaway.** DFCA converges at essentially the same rate as decentralized SGD, up to an additional term reflecting network connectivity. Crucially, all clients obtain all $k$ cluster models despite decentralized, asynchronous communication. The appendix provides a detailed proof by combining IFCA's cluster-assignment arguments with standard decentralized SGD analyses.

## 5 EXPERIMENTS

Next, we present our experiments with DFCA in practical learning settings. As common in the clustered FL literature (Ghosh et al., 2021; Lin et al., 2025; Ruan & Joe-Wong, 2022), we conduct experiments on the MNIST (Krizhevsky & Hinton, 2009), EMNIST (Cohen et al., 2017), CIFAR-10 (LeCun et al., 1998), and FEMNIST (Caldas et al., 2019) datasets, while applying rotations to the data to create different distributions. Our method is compared to the decentralized soft-clustering method FedSPD (Lin et al., 2025) and the optimized Decentralized Federated Averaging algorithm DFedAvgM Sun et al. (2021). IFCA (Ghosh et al., 2021) serves as the centralized baseline. After providing results for additional experiments with different connection probabilities, we discuss the communication efficiency and analyze the results of our experiments. Further details on the exact settings of our experiments can be found in the appendix.

### 5.1 DFCA EXCELS AMONG DECENTRALIZED APPROACHES

Our experiments demonstrate that DFCA consistently outperform the decentralized baselines Fed-SPD and DFedAvgM while achieving accuracy comparable to the centralized IFCA algorithm (Table 1). Figures 3a and 3b additionally show DFCA-GI converging at a similar rate as IFCA, while DFCA-LI converges slower but steeper than the other two methods. In more heterogeneous settings with larger numbers of clients (MNIST, EMNIST, FEMNIST; Table 2), DFCA maintains competitive performance, indicating that the sequential aggregation strategy effectively preserves cluster-specific models even as heterogeneity increases. IFCA's unusually high standard deviation for the EMNIST experiments with $k = 4$ occurs because IFCA detected only three clusters in one of its five runs.

Table 1: **DFCA outperforms other decentralized methods.** Experiments with EMNIST ($N = 100$ clients) and CIFAR-10 ($N = 50$ clients) show that both DFCA variants outperform existing DFL baselines while achieving comparable accuracy to the centralized baseline IFCA.

| | DFL | | | | CFL |
|---|---|---|---|---|---|
| Dataset | DFCA-GI | DFCA-LI | FedSPD | DFedAvgM | IFCA |
| MNIST ($k = 2$) | **93.7 ± 0.07** | 92.9 ± 0.06 | 86.2 ± 1.52 | 91.4 ± 0.21 | 93.9 ± 0.06 |
| EMNIST ($k = 2$) | **85.6 ± 0.13** | 85.3 ± 0.09 | 79.7 ± 0.92 | 73.5 ± 1.19 | 85.7 ± 0.11 |
| CIFAR-10 ($k = 2$) | **81.5 ± 0.40** | 80.4 ± 0.22 | 78.9 ± 0.23 | 76.0 ± 0.96 | 82.5 ± 0.11 |

**Insights.** In DFL, the way clients exchange model updates plays a crucial role in both convergence and efficiency. Beyond simple averaging, enabling clustered FL in decentralized networks

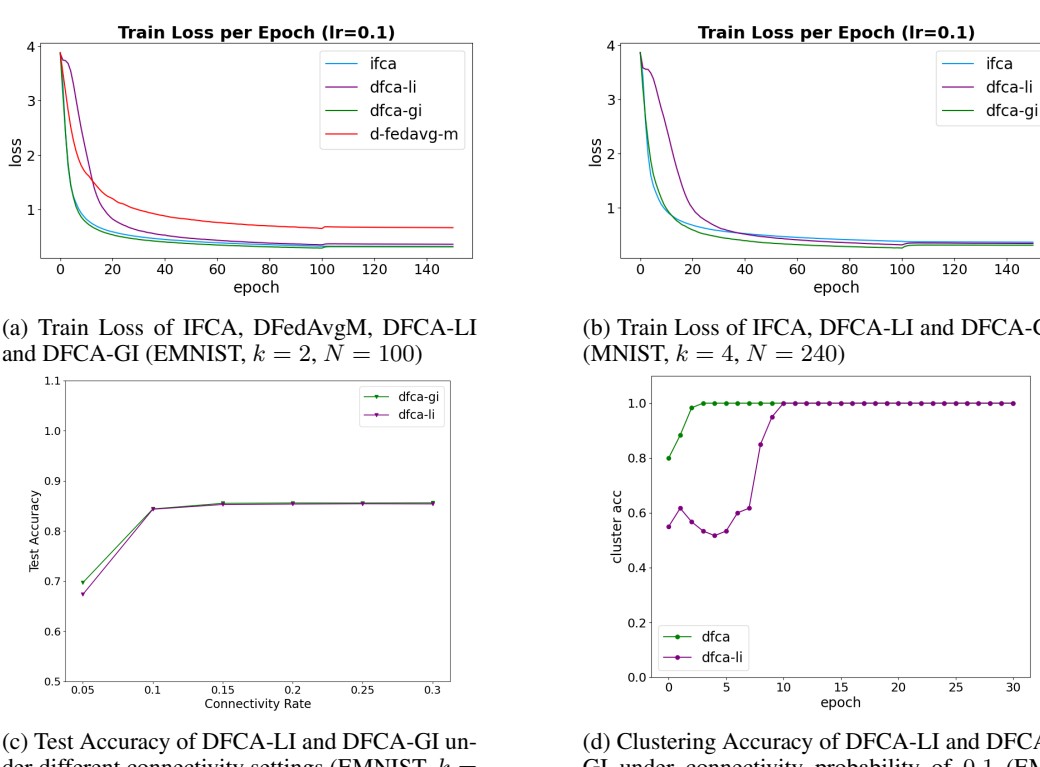

(a) Train Loss of IFCA, DFedAvgM, DFCA-LI and DFCA-GI (EMNIST, $k = 2$, $N = 100$)

(b) Train Loss of IFCA, DFCA-LI and DFCA-GI (MNIST, $k = 4$, $N = 240$)

(c) Test Accuracy of DFCA-LI and DFCA-GI under different connectivity settings (EMNIST, $k = 2$, $N = 100$)

(d) Clustering Accuracy of DFCA-LI and DFCA-GI under connectivity probability of 0.1 (EMNIST, $k = 2$, $N = 100$)

Figure 3: Graphs for MNIST/EMNIST experiments

Table 2: **DFCA is competitive with IFCA**. Additional comparisons with IFCA show that DFCA can perform on par within $1\%$ (mean) of IFCA's accuracy even in learning settings with higher heterogeneity ($N = 200$ for MNIST, $N = 100$ for EMNIST, and $N = 400$ for FEMNIST).

| | DFL | | CFL |
|---|---|---|---|
| Dataset | DFCA-GI | DFCA-LI | IFCA |
| MNIST ($k = 4$) | $92.8 \pm 0.63$ | $92.4 \pm 0.22$ | $93.1 \pm 0.73$ |
| EMNIST ($k = 4$) | $85.3 \pm 0.26$ | $85.1 \pm 0.20$ | $84.4 \pm 1.83$ |
| FEMNIST ($k = 4$) | $87.1 \pm 0.30$ | $86.4 \pm 0.15$ | $88.2 \pm 0.11$ |

is particularly valuable, as it allows clients with heterogeneous data to specialize in distinct model clusters without relying on a central coordinator. The general advantages of DFL, such as improved scalability, resilience to single points of failure, and better suitability for bandwidth-limited or peer-to-peer networks, have already been highlighted in prior works (Lalitha et al., 2018; 2019; Yuan et al., 2024). The results in Tables 1 and 2 show that DFCA not only outperforms the decentralized baselines but also does not fall short when compared to centralized IFCA. Despite evidence that DFL lags behind CFL (Sun et al., 2024), we reduce the accuracy difference to about 1% in CFL's favor, including in non-IID and low-connectivity settings.

## 5.2 DFCA IS ROBUST AGAINST LOW-CONNECTIVITY

Figure 3c shows the test accuracy of DFCA-LI and DFCA-GI under different, fixed connectivity settings on EMNIST. There, we can observe that a connectivity of $0.15$ is sufficient and the test accuracy does not change significantly when further increasing the connectivity rate. In settings with connectivity probabilities below $0.1$, DFCA-LI attains slightly lower accuracies than DFCA-GI, which can be attributed to its slower convergence caused by the additional time required for clustering, as seen in Figure 3d.

**Insights.** DFCA leverages a sequential running average to integrate neighbor updates efficiently, avoiding the need to store all incoming models and allowing updates to proceed asynchronously as they arrive. As a result, it improves scalability and robustness to network sparsity, while still achieving accuracy comparable to the centralized IFCA baseline.

# 6 RELATED WORK

## 6.1 DECENTRALIZED FEDERATED LEARNING

DFL originated from decentralized SGD optimization (Lian et al., 2018) and was later formulated by Lalitha et al. as a distinct concept for FL. During the following years, researchers proposed new frameworks and concepts around DFL, leading to rapid growth of the field. Research aspects of DFL include network topologies (Wang et al., 2019; Neglia et al., 2019; Malandrino & Chiasserini, 2021; Marfoq et al., 2020; Chellapandi et al., 2024), communication protocols (Sun et al., 2021; Lalitha et al., 2019; Hegedűs et al., 2019; Koloskova et al., 2019b; Hu et al., 2019; Bellet et al., 2018) and iteration orders (Yuan et al., 2024). Explicit DFL paradigms (Chang et al., 2018; Sheller et al., 2019; 2020; Huang et al., 2022; Yuan et al., 2023; Assran et al., 2019; Roy et al., 2019; Pappas et al., 2021; Shi et al., 2021; Chen et al., 2022; Wang et al., 2022) then put these concepts and assumptions in the context of real-world learning settings. However, there still remains a gap in performance between CFL and DFL (Sun et al., 2024), especially in low-connectivity settings and in the presence of heterogeneity, which motivated us to have a closer look into decentralized optimization.

## 6.2 CLUSTERED FL

First being introduced by Sattler et al. in 2019, clustered FL addresses the issue of handling heterogeneous data distributions of clients in a network. To optimize performance and adapt to different learning settings, researchers have introduced different methods to cluster the clients into groups with similar data distributions El-Rifai et al. (2025). After the initial introduction of client-side clustered FL algorithms based on client loss minimization Sattler et al. (2019); Ghosh et al. (2021), recent publications have focused on optimizing this strategy in different learning contexts (Mansour et al., 2020; Li et al., 2022; Kim et al., 2020). Voting-scheme-based (Gong et al., 2024) or k-means-based (Long et al., 2022) methods are alternative solutions utilizing client-side clustering. In contrast to the approaches mentioned above, our algorithm works in decentralized, low-connectivity settings without the need for a central instance. Lin et al. highlighted the potential of decentralized federated clustering methods when they introduced their decentralized soft-clustering algorithm for scenarios, in which clients possess multiple data distributions. However, FedSPD addresses a different scenario where each client may hold data from multiple distributions simultaneously (soft clustering), while our method assumes one data distribution per client (hard clustering). That motivated us to develop a decentralized approach capable of matching the performance of centralized IFCA in low-connectivity settings with clients holding different data distributions, as described in Chapter 2.

# 7 CONCLUSION

In this work, we introduced DFCA, a fully serverless method inspired by IFCA, that allows cluster-specific models to emerge and propagate through heterogeneous, sparse peer-to-peer networks. By employing a sequential running-average aggregation scheme, DFCA leverages stable learning with high clustering accuracy in heterogeneous environments where centralized methods are impractical.

Our experimental results demonstrate that DFCA achieves performance comparable to centralized IFCA while operating under decentralized communication constraints, and it consistently outperforms decentralized FedAvg with momentum and FedSPD. Furthermore, the sequential aggregation rule principally allows DFCA to operate asynchronously, making it well-suited for real-world networks with irregular connectivity and message delays.

Looking ahead, future work could investigate scaling DFCA to larger, non-IID datasets and analyzing the method with asynchronous model updates.

## USE OF LARGE LANGUAGE MODELS

Portions of this paper were prepared with the assistance of a large language model (LLM). In particular, we used an LLM to generate a first draft of the convergence proof for our proposed algorithm. The draft was then carefully checked, corrected, and verified by the authors before inclusion in the final manuscript. The LLM was also used to suggest stylistic edits and Latex formatting for improved readability. All conceptual contributions, experimental design, and validation of theoretical results were performed by the authors.

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

## A  APPENDIX

This manuscript benefited from language refinement and assistance in clarifying the convergence analysis using OpenAI's ChatGPT. All research ideas and contributions are solely those of the authors.

## A.1 EXPERIMENTAL SETTING

**EMNIST:** For the training with EMNIST (Cohen et al., 2017) (balanced split), we use $N = 100$ clients for $k = 2$ and $N = 200$ for $k = 4$ clusters and simulate two or four different data distributions by augmenting the datasets, applying 0, 180 or 0, 90, 180, 270 degree rotations to the data. The Convolutional Neural Network (CNN) used for training contains two convolutional layers, each followed by a relu activation function, a max-pool layer and a batch normalization layer. The models are trained for $\tau = 5$ local epochs with a learning rate of $\gamma = 0.1$, using Stochastic Gradient Descent over $T = 150$ global iterations. For the connection between clients, we use the adjacency matrix of an Erdős–Rényi graph with a connection probability of $0.15\%$. All experiments are run on five random seeds with the metric values being averaged over all runs.

**MNIST:** The training with MNIST (Krizhevsky & Hinton, 2009) is conducted on $N = 240$ clients and $k = 4$ clusters and data distributions (0, 90, 180, 270 degree rotations). We use a simple Multilayer Perceptron (MLP) with one hidden layer of size 2048 followed by a relu activation function. The other training parameters stay consistent with the EMNIST experimental setting, with the exception of reducing the connection probability to $0.1$.

**CIFAR-10:** The setup for our experiments with the CIFAR-10 (LeCun et al., 1998) dataset is similar to the EMNIST setup. We train with an identical CNN architecture over $N = 50$ clients and $k = 2$ clusters. We change the learning rate to $\gamma = 0.25$ and the graph connection probability to $0.2$.

**FEMNIST:** To test the algorithm in settings with even higher heterogeneity, we conducted experiments on the FEMNIST (Caldas et al., 2019) dataset. The training is done on $N = 400$ clients, who each get data from one distinct writer, with $k = 4$ clusters with a graph connection probability of $0.2$ and all other parameters equal to the MNIST experiments.

## B CONVERGENCE ANALYSIS

We provide a proof template that reuses standard ingredients from clustered FL (e.g., Ghosh et al. (2021)) for the assignment and from decentralized SGD/gossip (e.g., (Lian et al., 2017; Koloskova et al., 2019a; Boyd et al., 2006; Nedić & Olshevsky, 2016)) for communication. Throughout, expectations are with respect to the stochasticity of data sampling and any communication randomness.

**Notation.** Clients are $M = \{1, \ldots, N\}$, connected by an undirected graph $G = (M, E)$ with neighborhoods $\mathcal{N}_i$. The $k$ cluster index set is $[k] = \{1, \ldots, k\}$ and the (unknown) partition is $\{\mathcal{S}_1, \ldots, \mathcal{S}_k\}$ with data distributions $\{\mathcal{D}_1, \ldots, \mathcal{D}_k\}$. Client $i$ stores parameters $(\theta_{i,1}, \ldots, \theta_{i,k}) \in (\mathbb{R}^d)^k$. For cluster $j$, define the stacked vector $\Theta_j = (\theta_{1,j}, \ldots, \theta_{N,j})$ and the network average $\bar{\theta}_j^t = \frac{1}{N} \sum_{i=1}^{N} \theta_{i,j}^t$. The client loss is

$$F_{\text{client}}(\theta_{i,j}, D_i) = \frac{1}{|D_i|} \sum_{d \in D_i} \mathcal{L}(\theta_{i,j}, d), \quad F_{\text{cluster}}(j) = \sum_{i \in \mathcal{S}_j} F_{\text{client}}(\theta_{i,j}, D_i), \quad F_{\text{global}} = \sum_{j=1}^{k} F_{\text{cluster}}(j). \tag{14}$$

We measure per-cluster *disagreement* by $\text{Disp}_j^t = \frac{1}{N} \sum_{i=1}^{N} \|\theta_{i,j}^t - \bar{\theta}_j^t\|^2$.

**Lemma 1** (Assignment is descent for $F_{\text{global}}$)**.** *Conditioned on parameters* $\{\theta_{i,j}^t\}$, *the assignment step does not increase* $F_{global}$:

$$\sum_{i=1}^{N} \min_j F_{client}(\theta_{i,j}^t, D_i) \leq \sum_{i=1}^{N} F_{client}(\theta_{i,c_{t-1}(i)}^t, D_i). \tag{15}$$

*Proof.* Pointwise argmin over $j$ per client $i$ can only reduce the sum; cf. Ghosh et al. (2021). $\square$

**Lemma 2** (Local SGD descent with disagreement penalty)**.** *Let* $\gamma \leq 1/L$. *Then, conditioned on* $\Theta^t$,

$$\mathbb{E}\left[F_{cluster}(j; \bar{\theta}_j^{t+\frac{1}{2}}) \mid \Theta^t\right] \leq F_{cluster}(j; \bar{\theta}_j^t) - \frac{\gamma}{2} \|\nabla F_{cluster}(j; \bar{\theta}_j^t)\|^2 + \gamma^2 L\left(\sigma^2 + L^2 \text{Disp}_j^t\right). \tag{16}$$

*Proof.* Apply the smoothness descent lemma to the cluster-sum objective using unbiased gradients, and decompose the error into stochastic noise $\sigma^2$ and a consensus term proportional to $\text{Disp}_j^t$. This

form follows standard decentralized SGD analyses, e.g. Lian et al. (2017); Koloskova et al. (2019a). □

**Lemma 3** (Gossip preserves averages and contracts disagreement). *For each $j$, $\bar{\theta}_j^{t+1} = \bar{\theta}_j^{t+\frac{1}{2}}$. Moreover, in the synchronous (fixed $W$) case,*

$$\mathbb{E}\left[\operatorname{Disp}_j^{t+1} \mid \Theta^{t+\frac{1}{2}}\right] \leq \lambda^2 \operatorname{Disp}_j^{t+\frac{1}{2}}. \tag{17}$$

*In the async case, for some window $B$ and $\tilde{\lambda} \in (0, 1)$, $\mathbb{E}[\operatorname{Disp}_j^{t+B}] \leq \tilde{\lambda}^2 \operatorname{Disp}_j^t$.*

*Proof.* Average preservation follows from row-stochasticity (and doubly-stochasticity in the synchronous case). Disagreement evolution is governed by multiplication with $W_t^{(j)}$; contraction follows from the spectral gap (synchronous) or joint-connectivity arguments for randomized gossip (Boyd et al., 2006; Nedić & Olshevsky, 2016). □

**Lemma 4** (Assignment stabilization). *Under (A1)–(A5) with sufficiently small $\gamma$, there exists a finite $\tau$ such that $c_t(i) = c_\star(i)$ for all $i$ and all $t \geq \tau$.*

*Proof sketch.* By Lemmas 2–3, the averages $\{\bar{\theta}_j^t\}$ descend and the disagreements $\operatorname{Disp}_j^t$ contract, so all client copies tracking a fixed $j$ enter and remain in a neighborhood of $\theta_j^\star$. Within this neighborhood, separability (A5) enforces a unique, correct argmin, hence stable assignments; cf. Ghosh et al. (2021). □

**Theorem 2** (Convergence of DFCA). *Assume (A1)–(A5), choose $\gamma \leq c/L$ for a small numerical constant $c$, and let $\lambda$ (resp. $\tilde{\lambda}$) be the consensus factor in the synchronous (resp. async) case. Then:*

*(i)* *(Pre-stabilization) $F_{global}^t$ is non-increasing in expectation across assignment and local steps. The disagreements $\{\operatorname{Disp}_j^t\}$ remain bounded and contract at rate $\lambda$ (or $\tilde{\lambda}$ over windows).*

*(ii)* *(Stabilization) There exists $\tau < \infty$ such that $c_t(i) = c_\star(i)$ for all $t \geq \tau$.*

*(iii)* *(Post-stabilization) For $t \geq \tau$, DFCA is $k$ independent copies of decentralized SGD on $F_{cluster}(j)$.*

  • *Under (PL),*

$$\mathbb{E}\left[F_{global}^{\tau+T} - F_{global}^\star\right] \leq (1 - \mu\gamma/2)^T C_0 + O\left(\frac{\gamma\sigma^2}{\mu}\right) + O\left(\frac{\gamma L}{1-\lambda}\sigma^2\right), \tag{18}$$

  *with $C_0$ depending on the gap at $t = \tau$; in async, replace $(1 - \lambda)$ by the windowed $(1 - \tilde{\lambda})$.*

  • *Under (Cvx),*

$$\frac{1}{T}\sum_{t=\tau}^{\tau+T-1}\sum_{j=1}^k \mathbb{E}\|\nabla F_{cluster}(j; \bar{\theta}_j^t)\|^2 \leq O\left(\frac{F_{global}^\tau - F_{global}^\star}{\gamma T}\right) + O(\gamma L\sigma^2) + O\left(\frac{\gamma L}{1-\lambda}\sigma^2\right), \tag{19}$$

  *and choosing $\gamma = \Theta(1/\sqrt{T})$ yields the usual $O(1/\sqrt{T})$ rates (with the consensus penalty).*

*Proof sketch.* Combine Lemma 1 (assignment descent), Lemma 2 (SGD descent with a disagreement term), and Lemma 3 (average preservation and disagreement contraction). Lemma 4 yields finite-time stabilization, after which each cluster index $j$ follows a standard decentralized SGD recursion; apply known rates under PL or convexity and sum over $j$. □

**Remarks.** *(i) Batch vs. sequential aggregation.* The sequential "running average" update $\theta \leftarrow \frac{r}{r+1}\theta + \frac{1}{r+1}\theta_{\text{new}}$ implements a valid stochastic gossip step; the windowed contraction in Lemma 3 covers it. *(ii) Initialization.* Global initialization (DFCA-GI) sets $\operatorname{Disp}_j^0 = 0$ and typically reduces $\tau$; local initialization (DFCA-LI) only changes constants. *(iii) Clients not training $j$.* They still mix $\theta_{i,j}$ by applying $W_t^{(j)}$ to their current value; average preservation and contraction remain valid.

