# OpenReview forum: "DFCA: Decentralized Federated Clustering Algorithm"
_ICLR.cc/2026/Conference — ICLR 2026 Conference Withdrawn Submission_

### Official Review · Reviewer_1ZnZ · 2025-10-21

**Soundness:** 2
**Presentation:** 3
**Contribution:** 2
**Rating:** 2
**Confidence:** 4

**Summary:**

The authors present a new clustering method for decentralized FL extending the core idea of IFCA in absence of a central server. Each client stores all the cluster models and it is dynamically assigned to the cluster whose model minimizes the empirical loss. In the paper, the authors also present two possible aggregation strategies, a batch aggregation strategy and a sequential strategy that better meets the constraints of real world FL deployment. The paper is well motivated and quite easy to follow.

**Strengths:**

I appreciated the clarity of the paper. As mentioned above it is easy to follow and well motivated. In particular, the authors did provide a good theoretical extension of IFCA's proofs to the decentralized setting, properly applying interesting results of algebraic graph theory.

**Weaknesses:**

1. The DFL framework is interesting and I appreciate the effort of trying to extend IFCA to this scenario, however the similarities concerns me about the novelty of the proposed approach.

2. Storing all the cluster models on each client could be impractical, mostly in low-powered IoT frameworks where the storage capacity is often limited. I am concerned that in large scale scenarios  where the number of clusters drastically increases the memory cost of storing all the models explodes.

2a. The algorithm does not prevent from forming degenerate clusters, i.e. a cluster with a single client. In this case the model of the clusters coincides with the individual model of the client. Hence, this would imply that a client stores the individual information of another client, which is privacy concerning, with respect to standard FL privacy assumptions.

3. Since the number of clusters is fixed a priori as input of the algorithm, similarly to IFCA, why they evaluate the algorithms with $k = 4$? The evaluation shall be extended to other values of this hyper-parameter.

4. In the introduction the authors claim that their method is robust against data heterogeneity, however it is not precised with which value of Dirichlet's $\alpha$ they construct the federation.

5. I think that the authors should compare experimentally their method to other clustering DFL algorithms, or other DFL algorithms that are specially designed to mitigate data heterogeneity.

6. I advise the authors to update the literature review --- while most of the relevant historical works on CFL are present, the most recent literature is not properly discussed.


i. I do not get why the related work section has been placed at the end of the experimental results. It compromises the readability of the paper.

ii. In the pseudocode why the number of local SGD iterations equals the number of communication rounds?

**Questions:**

See weaknesses.

---

> ### Author Response · Authors · 2025-11-25
>
> Dear Reviewer,
>
> Thank you for your thorough assessment of strengths and weaknesses of the paper.
>
> **Weaknesses:**
>
> **1:** As we could not find any prior work introducing the same method. We would kindly ask the reviewer to point out the paper, which proposes our method? If the closeness to IFCA is the concern, our paper introduces several novel elements in comparison to IFCA: - Fully decentralized aggregation: Unlike IFCA, which relies on a centralized server for cluster updates, DFCA operates in a peer-to-peer network using synchronous or sequential gossip averaging.
> - Sequential running-average scheme: This design supports asynchronous updates and robust operation under sparse, low-connectivity networks, which is a practical extension not covered by IFCA.
> - Flexible initialization strategies: We explore both global and local initialization, allowing DFCA to operate effectively under heterogeneous starting conditions.
> - Empirical demonstration in decentralized settings: DFCA achieves accuracy comparable to centralized IFCA while scaling to decentralized networks, which requires non-trivial adaptation of the algorithm and analysis.
>
> **2:** We appreciate the reviewer’s concern regarding memory requirements in low-powered IoT devices. However, we do not see how this issue directly impacts the research question addressed in this work, which focuses on designing and evaluating a fully decentralized clustered FL algorithm. Addressing extreme memory constraints in large-scale IoT networks would require additional mechanisms (e.g., model compression or selective storage), which extend beyond the scope of the current paper.
>
> **2a:** The degenerate clusters or misclustering in general is a concern that can also be raised for other methods like IFCA. We analyzed that DFCA was more robust to this issue than IFCA, as i.e. seen in Chapter 5.1 (Note that misclusterings have been rather rare throughout our experiments). We are also going to add a discussion about misclusterings during the experiments to Section 5.
>
> To the privacy concerns: In decentralized federated learning, clients do not have global knowledge of cluster sizes, and each client only sees the models shared by its neighbors. Thus, there is no way for a client to know or infer how many clients belong to a given cluster. The algorithm never exposes individual data from other clients, at least not in the way described.
>
> **3:**  As reviewer xfoL raised the same concern, I will refer to the answer we gave: “For the decision of $k=4$ for the training on FEMNIST, we tested all methods with $k=10$ and observed that LI and GI, as well as IFCA only detected 4 clusters with significant amounts of clients. Choosing $k$ was based on a-priori knowledge about the number of data distributions in the case of CIFAR-10, EMNIST and MNIST, just as in IFCA.”
>
> **4:** In our experiments, we follow the same protocol as IFCA and generate heterogeneity via data rotation rather than Dirichlet partitions. We discussed this in the reply to reviewer xfoL: “Both baselines (IFCA and FedSPD) are evaluated on data rotation, as the setting is common and accepted in continual and federated learning research community, as originally motivated by Ghosh et al. 2020 (IFCA). Moreover, our experiments on FEMNIST introduce real naturally occurring data heterogeneity rather than synthetic data rotation. That said, we agree that experimental evaluation could be broadened further. We will extend the FEMNIST experiments, as well as other distributions of augmented data across clients, to show more diverse results.”
>
> **5:** As we did not find any decentralized, clustered FL methods that address the same data heterogeneity problem, could the reviewer please provide exact references? As other reviewers have also criticized the low number of baselines, we will try to extend the number of baselines to further display comparisons to other clustered FL algorithms designed for data heterogeneity.
>
> **6:** As mentioned in the previous paragraph, we could not find any decentralized, clustered FL methods, that seek the same goal as ours. Would the reviewer please specify which of the most recent literature is not included in this paper?
>
> **i:** Thank you for the feedback, we placed the related work section after the experiments to avoid breaking the flow of the methodological and experimental discussion. We will move it to section 2 of the paper to improve the readability.
>
> **ii:** Thank you for pointing us to this mistake in the pseudocode. It should be $\tau$, not $T$ and we corrected it accordingly.
>
> As the soundness score is at 2: Could the reviewer please advise us on how to improve the soundness of the paper?

---

### Official Review · Reviewer_DGbk · 2025-10-30

**Soundness:** 2
**Presentation:** 3
**Contribution:** 2
**Rating:** 2
**Confidence:** 4

**Summary:**

DFCA is a decentralized federated clustering algorithm inspired by IFCA, designed to mitigate data heterogeneity. It iterates through three steps. First, each client performs cluster assignment by locally evaluating all k cluster’s models on its data and self-assigning to the cluster with the minimum loss. Second, it executes a local update. Third, it conducts decentralized aggregation, where it exchanges and averages all k models with its neighbors, notably using a "sequential running average" to efficiently handle asynchronous updates.

**Strengths:**

1. DFCA achieves clustering in a DFL setting, and its performance is comparable to centralized IFCA under data-heterogeneous scenarios.
2. DFCA propose a practical sequential running average method to enable asynchronous aggregation. This mechanism is well-suited for realistic decentralized deployments .

**Weaknesses:**

1. The memory cost, computation cost, and communication cost are all large, each of which is k times higher than that of standard decentralized FL. First, DFCA requires each clients to store all k clustering models locally. Second, DFCA performs a complete inference on each of these k models using all their local data to find the model with the lowest loss. Third, the client needs to exchange all k models with its neighbors. The experiments in the paper are limited to small-scale settings such as k=2 and k=4, which masks the serious scalability issues of the design. If the number of clusters is slightly larger (such as k=10 or k=20), this k times memory and k times inference overhead is completely impractical for resource constrained devices.

2. The description of aggregation is contradictory. The aggregation defined by Eq. (6) and (7) is that client i only interacts with the neighbors N_ij in cluster j. However, the convergence analysis in Section 4 (Eq. 10) assumes that client i will interact with all neighbors N_i aggregate all k models. These two descriptions are completely inconsistent. The author must clarify which one is the true aggregation method of DFCA.

Overall, the methods proposed in this paper lack novelty and technical inspiration.

**Questions:**

see the above.

---

> ### Author Response · Authors · 2025-11-25
>
> Dear Reviewer,
>
> Thank you for your assessment of strengths and weaknesses of the paper.
>
> **Weaknesses:**
>
> **1:** We appreciate the concerns regarding memory, computation, and communication costs when the number of clusters $k$ grows. However, these costs scale linearly with $k$ by design, which is inherent to all client-side loss-based clustered FL methods, including IFCA. Our paper focuses on demonstrating the feasibility and accuracy of DFCA in decentralized networks rather than optimizing for extreme resource constraints. Addressing large-$k$ scenarios in memory- or compute-limited devices would require additional mechanisms such as model compression, selective storage, or hierarchical clustering, which extend beyond the scope of this work. Nonetheless, we ran experiments with $k=10$ for FEMNIST, where both IFCA and DFCA only detected 4 real clusters.  However, DFCA remains performant, showing that the algorithm can handle moderate $k$ values in practical settings without a-priori knowledge about the number of data distributions. Despite all that, we agree that the experiments should be more thorough and will add more detailed results to the appendix.
>
> **2:** Equation 10 uses stacked/matrix notation for compactness. Writing
>
> $\theta^{t+1}\_{i,j} = \sum\_{m \in \{i\} \cup \mathcal N\_i} w^{(j)}\_{im,t} \, \theta^{t+\frac12}\_{m,j}$
>
> is correct per cluster $j$, because $W_t^{(j)}$ is defined such that it only includes neighbors who contributed a model for cluster $j$.
>
> Operationally, each client does not receive all $k$ models from each neighbor, just the single model that neighbor updated. The matrix notation encodes this implicitly. For the theoretical analysis later in the chapter and in the appendix, we write $\Theta_j^{t+1} = W_t^{(j)} \Theta_j^t$.
>
>
> *Overall, the methods proposed in this paper lack novelty and technical inspiration.*
>
> We respectfully disagree with such a general characterization. While our method builds upon IFCA, the primary contribution of this paper is to extend the method to a fully decentralized setting with gossip-based aggregation with sequential averaging, as well as local and global initialization, which to our knowledge, has not been addressed in prior work and extends current research to a notably. The convergence analysis supports these contributions and complements the experimental validation.
>
> As the soundness score is at 2: Was this score caused by the points raised in **Weakness 2**? If not, we would appreciate advice on how to improve the soundness of the paper.

---

### Official Review · Reviewer_xfoL · 2025-11-01

**Soundness:** 3
**Presentation:** 2
**Contribution:** 2
**Rating:** 2
**Confidence:** 3

**Summary:**

This paper presents DFCA, a Decentralized Federated Clustering Algorithm that combines federated clustering (as in IFCA) with decentralized communication. The goal is to remove reliance on a central server and allow clients connected via a communication graph to collaboratively train cluster-specific models. To handle asynchronous communication, the authors propose a running average aggregation scheme and provide convergence analysis under smoothness and separability assumptions. Experiments on several benchmark datasets (MNIST, EMNIST, CIFAR-10, FEMNIST) suggest that DFCA can achieve accuracy comparable to centralized IFCA while outperforming other decentralized baselines.

**Strengths:**

1. The proposed approach is conceptually simple, intuitive, and easy to implement within decentralized federated learning frameworks.
2. While the theoretical contribution is not highly novel, the paper provides a useful convergence discussion that helps connect DFCA with existing decentralized SGD analyses under standard smoothness and connectivity assumptions.

**Weaknesses:**

1. The theoretical section mostly adapts existing results from IFCA and decentralized SGD. It does not introduce new analytical techniques or address the harder questions that arise from decentralization, e.g., how delayed neighbor updates or misclustered clients affect convergence.
2. The experimental evaluation in this paper is not sufficiently comprehensive. While standard datasets are used, the experiments focus on relatively small-scale and synthetic heterogeneity (via data rotation).
3. The paper does not isolate the contribution of its design choices, e.g., running average vs. synchronous aggregation, GI vs. LI initialization. Additionally, comparisons with more recent personalized or clustered FL methods (e.g., pFedMe, FedProx-based decentralized variants) are missing.
4. The algorithm requires each client to maintain $k$ full model copies and assumes stable connectivity across the network. This limits scalability to large $k$ or unstable peer-to-peer networks.

**Questions:**

1. How sensitive is DFCA to temporary misclusterings? Does the convergence argument still hold if clients frequently switch clusters or if cluster separability is weak?
2. How does the running average scheme compare empirically with simple synchronous averaging? Is there a measurable benefit in terms of wall-clock time or communication efficiency?
3. Have the authors evaluated DFCA on larger networks (e.g., $N>500$) or more realistic non-IID data partitions (e.g., natural label skew)? How does the method behave under dynamic connectivity or dropped messages?
4. How sensitive is performance to the number of clusters $k$ and to initialization (GI vs. LI)? Could the method degrade to standard decentralized SGD when $k=1$?
5. Could the authors clarify why comparisons with more recent personalized or clustered decentralized FL baselines (e.g., FedProx-Decentralized, Per-FedAvg) were not included?
6. Could the authors provide additional experiments to strengthen the evaluation? For example, have they considered testing DFCA on larger-scale or more realistic non-IID datasets, varying the number of clusters k, or evaluating its performance under strongly asynchronous or dynamically changing communication graphs?
7. Additionally, could the authors investigate communication and computation trade-offs, as well as scalability on larger networks?

---

> ### Author Response · Authors · 2025-11-25
>
> Dear Reviewer,
>
> Thank you for your thoughtful feedback and for raising critical points that help us improve the paper.
>
> **Weaknesses:**
>
> **1:** As stated in Chapter 1, our main contribution is the formulation of a decentralized federated clustering algorithm inspired by IFCA and designed with a sequential running-average, as well as experiments showing that DFCA matches or outperforms the accuracy of both the centralized and decentralized baselines. We added the formal proof to support our claims and experiments theoretically and to make this work rigorous and complete, not to extend the theoretical questions of decentralized optimization.
>
> **2:** Both baselines (IFCA and FedSPD) are evaluated on data rotation, as the setting is common and accepted in continual and federated learning research community, as originally motivated by Ghosh et al. 2020 (IFCA). Moreover, our experiments on FEMNIST introduce real naturally occurring data heterogeneity rather than synthetic data rotation. That said, we agree that experimental evaluation could be broadened further. We will extend the FEMNIST experiments, as well as other distributions of augmented data across clients, to show more diverse results.
>
> **3:** Our design choices, such as using a running average instead of synchronous aggregation and comparing GI versus LI, were selected to keep DFCA fully decentralized and communication efficient. While the experimental results demonstrate that these components work well together, we agree that the paper does not explicitly isolate their individual contributions. Due to the page limit these contributions could only be narrowly analyzed in the main part of the paper, but we acknowledge that experiments and an extensive analysis should be added to the appendix. That said, we will add some extensive experiments to the appendix including the points raised in **Weakness 2**.
>
> As for the other baselines, can you please point out other decentralized methods that attack the same problem? pFedMe is a personalized method from 2020 and does not inherently have the same goal as our algorithm, therefore we think including such methods in a comparison might not be fair. Personalized FL focuses on client-level specialization, aiming to learn a distinct model per client. In contrast, DFCA/IFCA aim to learn a small global set of models shared across all clients, such that every client maintains (and evaluates) all cluster models and the system as a whole becomes robust to diverse data distributions. Could the reviewer suggest an exact method for FedProx-based decentralized variants? But as other reviewers have also criticized the number of baselines, we will try to extend the number of baselines to further display comparisons to other clustered FL algorithms designed for data heterogeneity.
>
> **4:** This is not a weakness of our method specifically, but of client-sided training loss based clustering methods for Federated Learning in general (such as IFCA or FedSPD). That said, we also show that having only k=4 clusters helps boost the performance of collaborative training through our FEMNIST experiments, where every client receives data from a distinct writer, creating well over 4 different data distributions and higher feature skew. As mentioned in the previous paragraphs, we will update and extended these experiments and add them to the appendix.
>
> **Questions:**
>
> **1:** The results of our experiments showed that DFCA is more robust to misclusterings. In the experiments, one can see that clients normally only switch clusters in the first 20-30 epochs. This means that if misclusterings hold until the 30th epoch, it is likely that the client will remain in the wrong cluster throughout the whole training. In the case of our experiments, DFCA showed more robustness to those cases than IFCA, as we could see in the EMNIST experiments with $k=4$, where DFCA even outperforms IFCA on average. This point has been mentioned in chapter 5.1, but could be generalized in the main part or extended in the appendix.

---

> > ### Author Response · Authors · 2025-11-25
> >
> > **2:** The running-average scheme is mainly motivated by practical constraints of decentralized FL, where synchronous global aggregation is impractical or costly. Empirically, the difference is straightforward: synchronous averaging has global synchronization barriers, so that all clients must finish their local steps and communicate at the same time. Running averages allow fully asynchronous progress. In wall-clock terms, this removes idle waiting and reduces straggler effects, which typically leads to faster effective convergence per unit time even if the number of communication rounds is similar.  In communication terms, running averages reuse local, peer-to-peer updates and avoid the overhead of coordinating a global averaging round. Thus, while the per-update communication cost is similar, the overall scheme is more bandwidth-efficient in sparse or time-varying networks. Our paper does not isolate these effects experimentally, as it is well out of scope four our research question. However, the qualitative advantage is consistent with prior decentralized literature (e.g., gossip-based SGD), where asynchronous mixing yields lower wall-clock time under heterogeneous client speeds and connectivity.
> >
> > **3:**  We have tested performance on larger networks, i.e. with $N=1200$, which showed no difference in performance for EMNIST and MNIST. We will add this to the appendix in the later version. The motivation for DFCA, as in IFCA, is feature/covariate heterogeneity, where the optimal model parameters differ across client groups. In such settings, clustered models are well-justified, while under label-skew the multi-model formulation is unnecessary. We seek to research DFCA under more dynamic connectivity in future work, where we also want to address other communication related advantages and challenges.
> >
> > **4:** We have tested both GI and LI under $k=2$ and $k=4$ in the tables. For the decision of $k=4$ for the training on FEMNIST, we tested all methods with $k=10$ and observed that LI and GI, as well as IFCA clustered the clients into four groups, even with $N=400$. Choosing $k$ was based on a-priori knowledge about the number of data distributions in the case of CIFAR-10, EMNIST and MNIST, just as in IFCA.
> >
> > **5:** Acknowledged and discussed in **Weakness 3**.
> >
> > **6:** Does this question raise the same points as **Weakness 2**, **Question 3** and **Question 4**? If it is the case, this question  been answered in the previous paragraphs.
> >
> > **7:** Although this is an interesting point to investigate, this is a topic that is out of the scope of our research, especially with the page limit. This should be investigated in future research and incremental work.
> >
> >
> > As the presentation score is at 2: Could the reviewer please explain how we can improve the presentation of the paper?

---

### Official Review · Reviewer_t7yJ · 2025-11-03

**Soundness:** 3
**Presentation:** 3
**Contribution:** 2
**Rating:** 4
**Confidence:** 4

**Summary:**

This paper proposes DFCA (Decentralized Federated Clustering Algorithm), which, as the name suggests, finds clusters of federated learning clients in a decentralized manner while these clients train models on their local data. The key idea is to use a sequential running average of neighboring clients’ updated model parameters in order to compute model updates, instead of attempting to aggregate all neighbors’ parameter updates at once in each round. Such sequential averaging is naturally amenable to asynchronous updates, and the paper shows that it reduces to decentralized stochastic gradient descent (SGD) after the clients’ clusters converge, which is proven to happen after a sufficient number of update rounds. Experiments show that DFCA can nearly match the performance of centralized IFCA (Iterative Federated Clustering Algorithm), while outperforming other decentralized federated learning algorithms.

**Strengths:**

+ The experimental results show that DFCA outperforms several decentralized federated learning baselines, on multiple different datasets. It also nearly matches the accuracy achieved by IFCA, which is impressive for a decentralized learning algorithm.

+ The DFCA algorithm itself appears fairly easy to implement and can be naturally extended to asynchronous settings, which is especially important if clients’ network connections can change over time, which may not be synchronized across rounds.

**Weaknesses:**

--The client disagreement Disp_j^t is never formally defined, making it difficult to fully appreciate Theorem 1. Some assumptions are also unclear, e.g., wouldn’t the reduction to decentralized SGD rely on the fact that clients perform only one local training iteration between each update?

--It’s not clear why Assumption A3 has different graph mixing conditions for the synchronous and asynchronous cases. The conditions could also be defined more formally, e.g., what exactly does “disagreement contracts” mean? Wouldn’t that be a property of the algorithm as well as the underlying connectivity graph, not the graph mixing itself?

--The paper does not give many concrete examples of where DFCA might be deployed. For example, the assumption that clients are separated into distinct clusters appears fairly strong, so it would be useful to discuss some example applications where this assumption would be reasonable.

**Questions:**

1) There is no discussion of how (or whether) the proof of Theorem 1 differs significantly from prior work in the literature. My understanding is that the proof that the clusters eventually stabilize is fairly standard, though I’d appreciate clarification on this point from the authors (in particularly whether the sequential averaging makes a material difference to the proof technique).

2) Does DFCA assume that clients know their neighboring sets N_{I,j} at any given time? If not, how would they know when to resume assigning a new cluster to their data (going back to Step 1 of the algorithm) after the aggregation in Step 3 is complete? How would clients know this neighbor set in practice, since network connectivity may change over time?

3) The experiments section claims that DFCA is robust to low connectivity. I agree that the experimental results provide evidence of this claim, but is it evident in Theorem 1’s convergence result as well?

Please also see the questions listed in the weaknesses above.

---

> ### Author Response · Authors · 2025-11-25
>
> We sincerely appreciate your balanced feedback, pointing out the strengths and giving us advice for possible improvements of the paper.
>
> **Weaknesses:**
>
> **1:**
> The $Disp_j^t$ is formally defined in the introduction to the Convergence Summary in Chapter 5 (lines 270/271). We apologize for not emphasizing that part. As for the other point,  you are absolutely correct that Lian et al. assume only one epoch between each communication round. We explain that part in Lemma 2 of the appendix, but unfortunately cited the wrong paper of Koloskova et al., that extends the assumption of Lian et al. of one local SGD step between updates. In their 2020 Paper “A Unified Theory of Decentralized SGD with Changing Topology and Local Updates” (https://proceedings.mlr.press/v119/koloskova20a.html), they formulate the expected consensus rate with $\tau \geq 1$ (also formulated in Stich et al. 2019 (https://arxiv.org/pdf/1805.09767)). The extension to time-varying graphs is seen in Nedic & Olshevsky 2016, as cited in the proof reference in Lemma 3 of our paper and Chapter 5.3 of Koloskova et al.. We will update Lemma 2 to include an explanation of the difference in the literature for resubmissions.
>
> **2:** For the synchronous case, we have a sequence of mixing matrices $W_t$, that are row-stochastic and follow the graph. The only randomness is in the algorithm. That implies that we can impose the classical spectral contraction condition on the matrices themselves.
>
> For the asynchronous gossip case, mixing is driven by pairwise interactions, not deterministic matricies. That implies that the correct assumption is not on a matrix sequence but on the expected contraction of disagreement, which is the joint property of the graph (i), the activation distribution (ii) and the averaging rule (iii).
>
> That makes the two methods structurally different. This distinction is standard in e.g. Lian et al. 2017, Koloskova et al. 2020.
>
> The disagreement contracts are not a property of the algorithm but standard part of the graph mixing. As stated, they are formally defined by Lian et al. 2017, Koloskova et al. 2020.. We shortened that part to just a short explanation due to the page limit of 9 pages, but agree that formally defining them improves readability. We will add an extensive formulation of Assumption 3 for our future submission.
>
> **3:**  We agree that there should be a discussion section about possible cases, where DFCA can be deployed. However, another reviewer criticized making statements about possible applications without thoroughly testing and proving applicability. Therefore, we thought that extending this might blow the scope of this paper out of proportion. A resubmission will include a short discussion paragraph at the end of paper, discussing both applications, limitations and future work, as the testing and deeper analysis of applications should be researched in incremental work.
>
> **Questions:**
>
> **1:**  We can clarify that Theorem 1 mainly uses existing techniques of clustered federated learning and decentralized optimization to prove that DFCA performs within the theoretical frame of comparable algorithms. We basically additionally show that our techniques do not fundamentally change the argument of IFCA, but only affects the disagreement contraction and convergence rate of the decentralized SGD post-stabilization. We also want to clarify that a theoretical, formal extension of clustered federated learning is not the main research focus of this paper. The main contributions are listed in Chapter 1. We added the formal proof to support our claims and experiments theoretically and to make this work rigorous and complete.
>
> **2:** DFCA assumes each client to know its current neighborhood $\mathcal{N}\_i$ at the time of communication, but not necessarily cluster-specific subsets $\mathcal{N}\_{i,j}$ in advance. In the asynchronous case, the cluster of the exchange partner is only known at the time of model reception. In the synchronous case, the elements of $\mathcal{N}\_{i,j}$ are collected with each communication step and a formal set for the set of neighbors of each cluster for notation. Practically, the client would only need to store the set of models for each cluster for aggregation, not the set of clients. Step 1 does not require knowledge of all neighbors or their cluster assignments in advance, it is performed locally once a client has completed Step 3. In practice, clients only know which neighbors are currently reachable in the network.

---

> > ### Author Response · Authors · 2025-11-25
> >
> > **3:** Not directly. The convergence bound of Theorem 1 does include a consensus/graph contraction term via $\lambda$ (or $\tilde \lambda$ in the asynchronous case) formally capturing the effect of the connectivity graph on disagreement among clients. A graph with lower connectivity corresponds to a smaller spectral gap for larger $\lambda$, which slows down the contraction of disagreement and can increase the constant in the convergence bound. So while Theorem 1 does account for network topology, it does not automatically prove robustness to low-connectivity, it only proves convergence if the graph is connected. The experimental results do show that the algorithm is more robust than proven in the worst-case theoretical bound.

---

### Note · Authors · 2025-12-10

I have read and agree with the venue's withdrawal policy on behalf of myself and my co-authors.